# Detection of Trypanosoma Infection in Dromedary Camels by Using Different Diagnostic Techniques in Northern Oman

**DOI:** 10.3390/ani12111348

**Published:** 2022-05-25

**Authors:** Amal Al-Kharusi, Elshafie Ibrahim Elshafie, Senan Baqir, Asim Faraz, Aliya Al-Ansari, Pamela Burger, Osman Mahgoub, Kaadhia Al-Kharousi, Halima Al-Duhli, Mohammed Al-Sinani, Raqiya Al-Hatali, Derek Roberts

**Affiliations:** 1Ministry of Agriculture, Fisheries Wealth and Water Resources, Muscat 100, Oman; hhh.live89@gmail.com (H.A.-D.); sur.1992@live.com (M.A.-S.); rqya.88@gmail.com (R.A.-H.); 2Department of Animal and Veterinary Sciences, College of Agriculture and Marine Sciences, Sultan Qaboos University, Muscat 100, Oman; eielshafie@squ.edu.om (E.I.E.); osmahgob@hotmail.com (O.M.); kaadhia@squ.edu.om (K.A.-K.); 3Central Veterinary Research Laboratory, Al Amarat 8076, Sudan; 4Department of Biology, College of Sciences, Sultan Qaboos University, Muscat 100, Oman; baqirs@squ.edu.om (S.B.); alansari@squ.edu.om (A.A.-A.); derek75roberts@gmail.com (D.R.); 5Department of Livestock and Poultry Production, Bahauddin Zakariya University, Multan 60000, Pakistan; drasimfaraz@bzu.edu.pk; 6Research Institute of Wildlife Ecology, Department of Interdisciplinary Life Sciences, University of Veterinary Medicine Vienna, 1160 Vienna, Austria; pamela.burger@vetmeduni.ac.at

**Keywords:** camel, blood, trypanosomes, prevalence, sequences

## Abstract

**Simple Summary:**

The camel is a salient character of the pastoral economy around the world. Apropos of this title role, the camel plays a particularly crucial role in arid and semi-arid areas of the world. The camel population is growing despite much urbanization around the world. The importance of the camel concerns the acquisition of milk, meat and other byproducts. Due to emerging health issues, people mainly rely on camel milk and meat due to their remedial purposes, hence why it is held in high regard. Therefore, it is crucial to take into consideration the camels’ health, risks, associated diseases and diagnoses. In this investigation, an effort was made to detect trypanosomoses in dromedary camels by using different diagnostic techniques in Northern Oman.

**Abstract:**

Camel trypanosomoses is considered a devastating disease with severe health consequences that can be caused by different hemoprotozoan parasites. Camel samples (388) from the five regions in Northern Oman were assessed using a thin blood film. In addition, 95 seropositive samples were analyzed using various primers of mechanically transmitted trypanosomes. Out of the 388 blood smears examined, 0.8% (CI 95%, 2/388) were found to be positive for *Trypanosoma* sp. using a microscope. The parasitologically positive cases were detected in samples from females. The overall molecular prevalences were as follows: TBR was 78/95, 77% (CI 73.1–89.2%); ITS was 30/95, 31.6% (CI 73.1–89.2%); and *T. evansi* type A (RoTat 1.2) was 8/95, 8.4% (CI 4.0–16.0%). There were two species of trypanosomes that were observed in the camels.

## 1. Introduction

Trypanosomoses cause specific problems in camel production and lead to economic losses. The disease is caused by various species of trypanosomes: *T. brucei, T. congolense* and *T. vivax* [1]. *T. evansi* is the most important species due to having a wide host range of animals, such as camel, horse, cattle, buffalo, goat, sheep, dog, pig, tiger and Asian elephant, and causing vaccination failure such that animals will be more susceptible to having a disease [2,3]. In India, they found that humans were susceptible to trypanosomiases [4,5,6]. Parasitological examinations are not sensitive but can be used in the field with a low amount of equipment. To increase the sensitivity of the method, a hematocrit centrifuge technique or a buffy coat method are used [7]. Molecular methods are superior to antigen detection methods because they can detect pre-patent and chronic infections [8]. There are different markers that are used to detect and study species of trypanosomes. The internal transcribed spacer one (ITS1) and internal transcribed spacer two (ITS2) are valuable in the more discrete phylogenetic separation of piroplasms and their subspecies [9,10]. The predominant variant antigen type for the variable surface glycoprotein of trypanosomes RoTat 1.2 VSG is expressed in *T. evansi* [11]. The PCR RoTat 1.2 is used for *T. evansi* type A and PCR EVAB is used for *T. evansi* type B. Previous studies that reported on trypanosomes in Oman did not identify the trypanosome species using a molecular method. This study aimed to detect the prevalence of *Trypanosoma* in five regions in northern Oman using light microscopy and molecular diagnoses using four primers. 

## 2. Materials and Methods

### 2.1. Study Area and Sampling Collection

The study was conducted in five regions in the northern part of the Sultanate of Oman. A total of 388 samples from camels of different ages and sexes were collected from January to February 2017: from Al Buraimi (*n* = 31), Ad Dakhiliyah (*n* = 49), north and south Al Batinah (*n* = 86), north and south Ash Sharqiyah (*n* = 192), and Al Dhahirah (*n* = 30), with an expected prevalence of 50% and a 95% confidence interval. Thursfield (2007) calculated the minimum sample size required according to the following formula: *n* = 1.96^2^ × Pexp (1 − Pexp)/d^2^(1)
where *n* is the sample size; Pexp is the minimum expected prevalence, which is 50%; 1.96 is the value of z for the 95% confidence interval; and d is the desired accuracy level for the 95% interval. A total of 95 samples were seropositive from the 388 samples that were examined for molecular tests.

On the day of sampling, animal information such as age and gender were collected, along with the sample location. After the willingness of the camel’s owner to participate in the study was confirmed, 10 mL of blood from each animal was collected from the jugular vein in an ethylene diamine tetraacetic acid (EDTA) vacationer. The samples were put in a cool box and transferred to the laboratory for diagnostic procedures.

### 2.2. Examination of Trypanosome Using a Thin Peripheral Blood Smear 

A thin blood smear was prepared on a clean glass slide from freshly collected blood (collect within 24 h) and then stained using Wright’s stain (quick diff stain). The stained smears were allowed to dry and examined under a light microscope using an oil lens (BX53, OLYMPUS, Tokyo, Japan).

### 2.3. Molecular Analysis

Four molecular tests were used to detect *Trypanosoma evansi* in the camels. There were comparative studies that recommended which TBR primers were more sensitive for detecting *T. evansi* [12]. The PCR RoTat 1.2 was used for *T. evansi* type A and PCR EVAB was used for *T. evansi* type B [13,14,15]. 

DNA was purified using the E.Z.N.A. SQ blood DNA kit (Omega Bio-Tek, Inc., Norcross, GA, USA). The extracted DNA (300 µL) was quantified using a Nano Drop spectrophotometer 2000c (Thermo Scientific, Waltham, MA, USA) at a wavelength of 260/280 nm and was stored at −20 °C until used. 

The VSG RoTat 1.2 rDNA gene fragments were amplified using RoTat 1.2-F (GCGGGGTGTTTAAAGCAATA) and RoTat 1.2-R (ATTAGTGCTGCGTGTGTTCG) and they were given a 205 bp amplification product [13]. The polymerase chain reaction (PCR) contained 15 μL of master mix (Thermo Scientific, Fremont, CI, United States), 5 μL of nuclease-free water, 1 μL of F-primer and 1 μL of R-primer, and 3 μL of DNA template. The RoTat 1.2 PCR reaction was started in 25 μL at 95 °C for 10 min. The 40 cycles were initiated with denaturation for 30 s at 95 °C, followed by annealing for 45 s at 59 °C and extension for 30 s at 72 °C. The final extension was carried out for 7 min at 72 °C.

The minicircle rDNA gene fragments were amplified using EVAB-F (ACAGTCCGAGAGATAGAG) and EVAB-R (CTGTACTCTACATCTACCTC) primers Njiru et al. (2006) [15]. The EVAB PCR reaction was started in 25 μL at 95 °C for 10 min. The 40 cycles were initiated with denaturation for 30 s at 95 °C, followed by annealing for 45 s at 60 °C and extension for 30 s at 72 °C. The final extension was carried out for 7 min at 72 °C. The *ITS-1* gene fragments were amplified using ITS-1 F (TGTAGGTGAACCTGCAGCTGGATC) and ITS-2 R (CCAAGTCATCCATCGCGACACGTT) Fikru et al. (2012) [16]. The ITS PCR reaction was started in 25 μL at 95 °C for 10 min. The 40 cycles were initiated with denaturation for 30 s at 95 °C, followed by annealing for 45 s at 65 °C and extension for 30 s at 72 °C. The final extension was carried out for 7 min at 72 °C. It was given an approximately 480 bp amplification product.

The *ITS-1* gene fragments were amplified using TBR-1 F (GAATATTAAACAATGCGCAG) and TBR-2 R (CCATTTATTAGCTTTGTTGC), and they were given an 177 bp amplification product Masiga et al. (1992) [17]. The TBR PCR reaction was started in 25 μL at 95 °C for 10 min. The 40 cycles were initiated with denaturation for 30 s at 95 °C, followed by annealing for 45 s at 53 °C and extension for 30 s at 72 °C. The final extension was carried out for 7 min at 72 °C. The PCR product was checked in 1.5% agarose gel stained with a safe sybr DNA gel (Invitrogen 10X TAE Buffer; Vilnius, Lithuania). The band size was determined using a well loaded with a 1 kb ladder (Invitrogen; Vilnius, Lithuania).

### 2.4. Cleaning up the PCR Product

The PCR product was purified in 96-well plates using ethanol (Riedel de Haen; Munich, Germany). Approximately 60 µL of (100%) absolute ethanol was added to each well that contained the product of a sequencing reaction and was incubated at room temperature. After the mixture centrifugation at 1008 RCF for 30 min, 70 μL of 70% ethanol was added and centrifuged again. Then, 10 µL of formamide was added to each well and the plate was spun. The plate was incubated in PCR at 95 °C for 5 min. The reaction was done by using a Big Dye buffer (Thermo Fisher Scientific, Vienna, Austria). Mixture one was prepared, which contained 1.5 µL of Exosap and 3 µL of purified PCR product. The mixture was incubated at 37 °C for 15 min. The reaction was stopped by heating the mixture at 85 °C for 15 °C.

### 2.5. Sequence Reaction

Mixture two was prepared by provide the master forward and reverse solution by adding 1.5 µL of big dye Terminator v 3.1 (Thermo Fisher Scientific, Vienna, Austria), 3.5 μL big dye buffer, 2 µL of nuclease-free water and 1 µL of primer, and was added to mixture one. The 35 cycles were started with denaturation for 10 s at 96 °C, annealing for 5 s at 55 °C and extension for 4 min at 60 °C. The purified DNA was sequenced in a 3130×1 Genetic Analyzer (Applied Biosystems 3100, Norwalk, CT, USA).

### 2.6. Bioinformatics

Results from sequencing were edited using Bioedit version 7.0.5 [18]. The results were blasted against the trypanosome non-redundant sequences in the National Center for Biotechnology Information (NCBI) nucleotide database. The phylogenetic tree was generated from sequence alignments by using the MEGA X version 10 [19].

### 2.7. Statistical Analysis

The prevalence and the confidence interval (CI) for the proportion of camels in five regions in Oman using (TBR, RoTat 1.2, ITS) PCR detection of *Trypanosoma* was conducted with a 97% confidence level. The number of positive cases of trypanosomoses was mapped using the QGIS version 2 (Free software foundation, Inc., Boston, MA, USA, 1991).

## 3. Results 

In this study, 388 samples were examined from five regions: Al Buraimi (31), Ad Dakhiliyah (49), Al Batinah (86), Ash Sharqiyah (192) and Al Dhahirah (30). Out of the 388 blood smears examined, 0.8% (2/388) were found to be positive for trypanosomoses using a microscope. The parasitologically positive cases were reported from Ibra in Ash Sharqiyah. They were samples from females (Figure 1).

The extracted DNA of 95 seropositive samples was amplified using five primers (RoTat 1.2 -F and -R, EVAB, ITS-1, TBR). *T. evansi* Type B was not detected in any of the samples from the five regions. The overall molecular prevalence of *T. evansi* type A was 8.4%. The molecular prevalence of ITS-1 was 31.6%. The overall molecular prevalence of TBR was 77%. 

The overall molecular prevalence of TBR was 78/95 or 77% (CI 73.1–89.2%), with 100% (CI 16.0–100.0%) in Al Buraimi, 82.4% (CI 57.0–92.2%) in Al Batinah, 75% (CI 35.0–97.0%) in Ad Dakhiliyah, 79.2% (CI 66.0–89.2%) in Ash Sharqiyah and 77.0% (CI 73.0–89.2%) in Al Dhahirah (Figure 2). The overall molecular prevalence of ITS was 30/95 or 31.6% (CI 73.1–89.2%), with 100% (CI 16.0–100.0) in Al Buraimi, 23.5% (CI 6.8–49.9%) in Al Batinah, 35.8% (CI 23.1–50.2%) in Ash Sharqiyah and 40% (CI 16.3–67.7%) in Al Dhahirah (Figure 3). The overall molecular prevalence of *T. evansi* type A (RoTat 1.2) was 8/95 or 8.4% (CI 4.0–16.0%), with 50% (CI 2.0–99.0) in Al Buraimi, 23.6% (CI 7.0–50.0) in Al Batinah, 12.5% (CI 0.3–52.7%) in Ad Dakhiliyah and 4% (CI 0.5–13.1%) in Ash Sharqiyah (Table 1).

Thirty *Trypanosoma*-positive samples were sequenced with a yield of only six. Four haplotypes were identified. One trypanosome-positive sample (GenBank: MH247175) sequenced shared 100% identity with *T. evansi.* A trypanosome-positive sample (GenBank: AB551920) sequenced shared 100% identity with *T. evansi evansi*, one sequence shared 100% with *T. evansi evansi* and three samples were identical (GenBank: JN896755, KU552351), sharing 100% identity with *T. evansi* (Table 2). The strain that was detected in one sample (GenBank: AB551920) was the same as the one identified in Egypt. In addition, one haplotype (GenBank: FJ712716) was identical to the strain of *T. evansi evansi* found in camels in China [20]. Moreover, one dromedary camel sample had the same strain of *T. evansi* (GenBank: MH247175) that was identified in a *Stomoxys calcitrans* fly in Kenya [21]. Three camel samples had the same haplotypes; they were identical to strains detected in Iran (GenBank: JN896755) and were identical to a strain identified in blood samples of infected mice (GenBank: KU552351) [22,23]. 

The phylogenetic tree based on the sequence of the ITS-1 region indicated that the *T. evansi* obtained was related to the *T. evansi* detected in camels from Egypt (AB551920), China (FJ712716), Iran (JN896755) and Kenya (MH247175) (Figure 4, Table 2).

Eight positive samples were sequenced, and two haplotypes (alleles) were identified. The nucleotide sequence showed 100% identity with an isolated gene from camels in India (GenBank: KY457409.1, KY457408.1). Furthermore, they were closely related to *T. evansi* that was identified in cattle from Malaysia (GenBank: MT514514.1, MT514513) (Table 3).

## 4. Discussion

The prevalence observed using light microscopy was much lower (0.8%) than the molecular prevalence (8.4–77%). This outcome was expected due to the higher sensitivity of the molecular analysis. There were many studies that recorded the prevalence of trypanosomoses in dromedary camels using microscopic examinations. In Oman, two studies reported a higher prevalence than this study (44% and 2.6% in [24,25], respectively). In other countries, the prevalence of *Trypanosoma* species using microscopic detection were 0.8% [26] and 30.9% [27] in Saudi Arabia; 1.7% in Kuwait [28], 13% in Iran [29], 14% in Algeria [30]; 4.25% in Egypt [31]; 4.4% in Ethiopia [32]; 5.3% in Kenya [15]; 33% in Jordan [33]; 14.1%–1.7% in Sudan [34,35]; 6.55%–13.7% in Pakistan [36,37]; and 30% in Palestine [38]. In Somalia, the prevalence of *Trypanosoma* using microscopic examination was negative but positive using PCR [39]. The variation in the prevalence amongst countries around the world exists due to variations in the climate, hosts and blood samplings. Trypanosomoses is a known chronic disease; therefore, trypanosome species will be low in blood circulation. Thus, there is a lower chance of detecting the parasite via microscopy. Moreover, the number of circulating parasites in young camels is lower than in older camels [40,41]. This is because the immunity of older camels is low and their movement between villages is greater, which leads to an increase in the chance to detect the parasite in the blood. 

The prevalence of trypanosome species using ITS-1 PCR was higher in Al Buraimi (50%) governorates and lower in the Al Batinah governorates (23.5%). The overall prevalence was 31.6% in five governorates. Our results were lower than that detected in Sudan, which was 35.6% [39]. In Somalia, 5 out of 182 (2.7%, 95% Cl: 0.9–6.3%) camels were found to be positive for trypanosome species using ITS-1 PCR [41]. (Figure 5) The results showed that camels in the infected regions were positive for trypanosomosis, which was due to *T. evansi.* The detection via ITS1 PCR was a useful diagnostic method to distinguish *T. evansi* from other trypanosome varieties [42].

The prevalence of trypanosome species using TBR PCR was higher in Al Buraimi, followed by Al Dhahirah and Al Batinah. The overall prevalence was 77% in five governorates and was higher than the overall prevalence of trypanosoma species found using TBR in Pakistan: 31.9% [8] and 22% [37]. 

In this study, we detected *T. evansi* type A in eight samples from Al Buraimi, Al Batinah, Ad Dakhiliyah and Ash Sharqiyah. Here, we report *T. evansi* type A for the first time in Oman. Additionally, *T. evansi* type A was detected in camels from Sudan [42] and Egypt [43]. Furthermore, *trypanosome. evansi* type A was detected in camel, cattle, sheep, goat and donkey in Ethiopia [44] and Pakistan [37].

With EVAB PCR, *T. evansi* type B was not detected in any governorate. *Trypanasoma evansi* type B was isolated in Sudan, Kenya and Ethiopia [44,45,46,47]. This type occurs more in African countries [41]. This study had some limitations. The number of samples should increase, and in the future, we must screen all samples by using TBR primers and then positive samples should be examined to detect *T. evansi type A* and *T. evansi type B.* In future studies, more samples should be sequenced to obtain more details for the phylogenic analysis. 

## 5. Conclusions

Trypanosomiases is a disease that causes a reduction in camel production and has direct and indirect effects. To our knowledge, this study is the first to report *Trypanosoma evansi* infection in dromedary camels in Oman using a combination of parasitological and molecular diagnostic analyses, which are recommended by the OIE. *T. evansi* is not the specific species that was observed in camels in other countries, such as Iran and Sudan, where they observed the *Trypanosoma vivax* species. Additionally, it is recommended to expand the investigation to various animal species, such as horse, cattle, goat and sheep.

## Figures and Tables

**Figure 1 animals-12-01348-f001:**
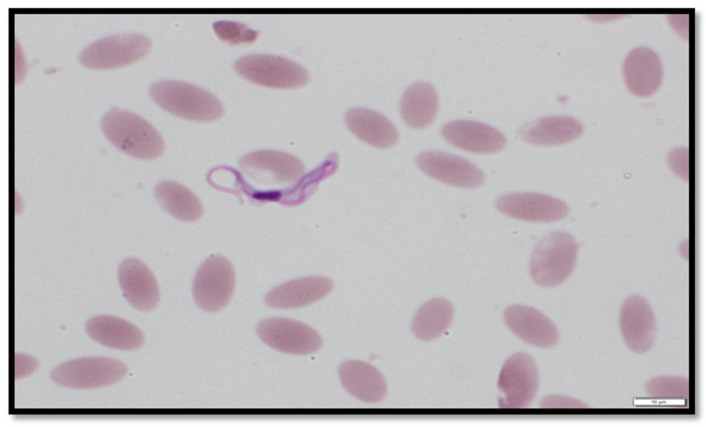
*Trypanosoma evansi* under a microscope (Olympus BX53, Tokyo, Japan).

**Figure 2 animals-12-01348-f002:**
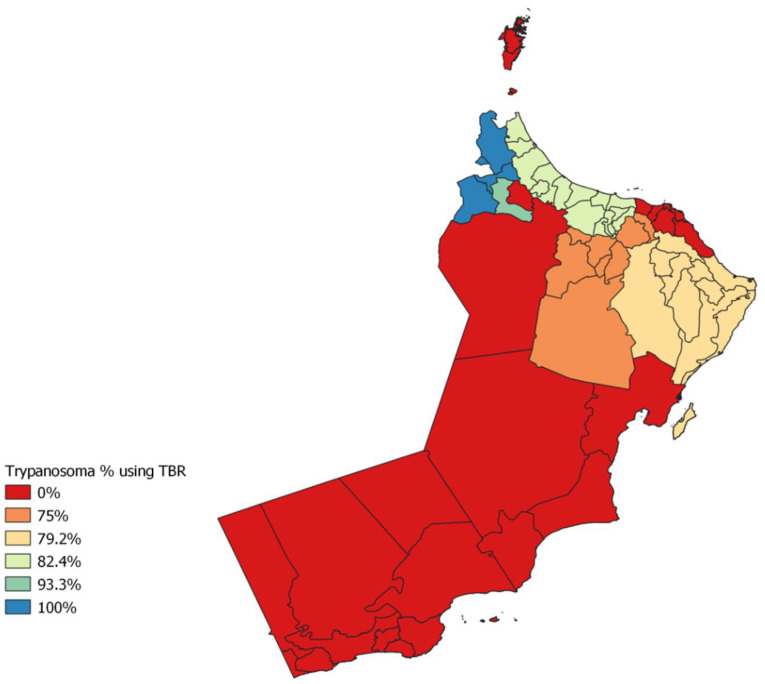
Distribution of trypanosomoses in camels in Oman by using TBR (QGIS 7.4).

**Figure 3 animals-12-01348-f003:**
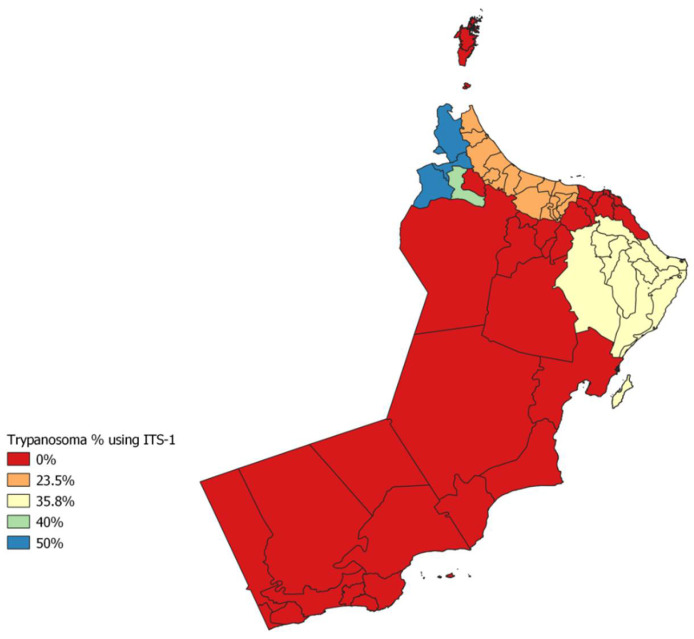
Distribution of trypanosomoses in camels in Oman by using ITS-1 (QGIS 7.4).

**Figure 4 animals-12-01348-f004:**
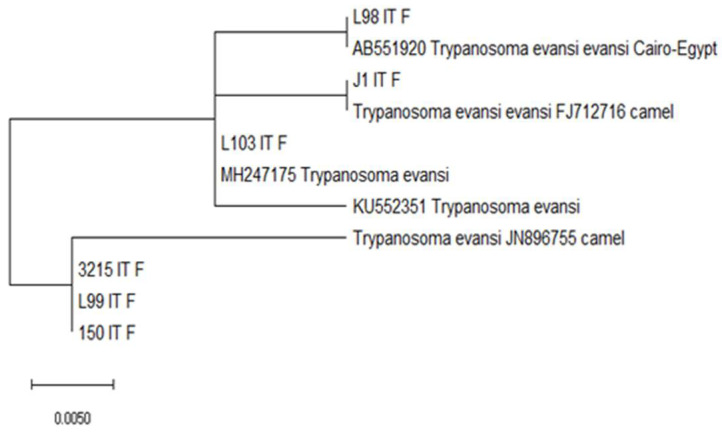
Phylogenetic tree of *T. evansi* ITS-1 sequences of alignment.

**Figure 5 animals-12-01348-f005:**
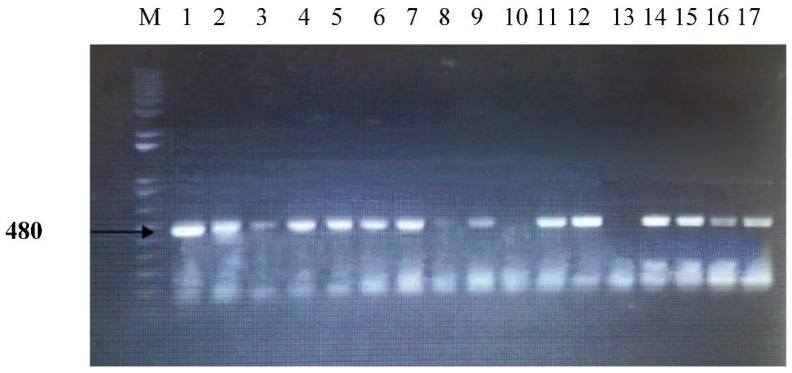
PCR products of the ITS-1 in an agarose gel electrophoresis M1 100 bp marker, lanes 1 to 7, 9, 11 to 12 and 14 to 17 *T. evansi* from positive samples in different locations.

**Table 1 animals-12-01348-t001:** The prevalence and the confidence interval (CI for the proportion of camels in five regions in Oman using (TBR, RoTat 1.2, ITS) PCR detection of *Trypanosoma*.

Governorate	Primer	Number of Samples	Prevalence (%)	95% Cl for Prevalence	
				Lower	Upper
Al Batinah	ITS-1	17	4/17 (23.5)	6.8	49.9
Ad Dakhiliyah		8	0/8 (0)	47.4	36.9 ^*^
Ash Sharqiyah		53	19/53 (35.8)	23.1	50.2
Al Buraimi		2	1/2 (50.0)	1.7	98.8
Al Dhahirah		15	6/15 (40)	16.3	67.7
Total		95	30/95 (31.6)	22.4	41.9
Al Batinah	TBR	17	14/17 (82.4)	57.0	92.2
Ad Dakhiliyah		8	6/8 (75.0)	35.0	97.0
Ash Sharqiyah		53	42/53 (79.2)	66.0	89.2
Al Buraimi		2	2/2 (100.0)	16.0 ^*^	100.0
Al Dhahirah		15	14/15 (93.3)	68.1	99.8
Total		95	78/95 (77.0)	73.0	89.2
Al Batinah	RoTat 1.2	17	4/17 (23.6)	7.0	50.0
Ad Dakhiliyah		8	1/8 (12.5)	0.3	52.7
Ash Sharqiyah		53	2/53 (4.0)	0.5	13.1
Al Buraimi		2	1/2 (50.0)	2.0	99.0
Al Dhahirah		15	0/15 (0)	0	22.0 ^*^
Total		95	8/95 (8.4)	4.0	16.0

* One-sided 97.5% CI approximation of the proportion.

**Table 2 animals-12-01348-t002:** Parasite species isolated from camels using ITS-1.

Sample No.	Primer	Parasite Species	Host	Accession Number	Identity
L103	ITS-1	*Trypanosoma evansi*	Whole fly	MH247175	100%
L98	ITS-1	*Trypanosoma evansi evansi*	Camel	AB551920	100%
J1	ITS-1	*Trypanosoma evansi evansi*	Camel	FJ712716	100%
3215	ITS-1	*Trypanosoma evansi*	Camel	JN896755	100%
L99					
150					
3215	ITS-1	*Trypanosoma evansi*	Camel	KU552351	100%
L99					
150					

**Table 3 animals-12-01348-t003:** Parasite species isolated from camels using RoTat 1.2.

Sample No.	Primer	Parasite Species	Accession Number	Host	Identity
150	RoTat 1.2	*Trypanosoma evansi*	KY457409.1	-	100%
3215		*Trypanosoma evansi*	KY457408.1	-	100%
		*Trypanosoma evansi*	MT514514.1	Cattle	100%
		*Trypanosoma evansi*	MT514513.1	Cattle	100%

## Data Availability

All the relevant data is available in this paper.

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
