# Peer review of "Detection of Trypanosoma Infection in Dromedary Camels by Using Different Diagnostic Techniques in Northern Oman"

_animals, 2022, doi:10.3390/ani12111348_

Round 1

Reviewer 1 Report

Manuscript Number: animals-1663480

Title: Detection of Trypanosomoses in Dromedary Camels by using different diagnostic techniques in Northern Oman

This study presents results of an epidemiological research on camels trypanosomosis in Oman. Authors detected T. evansi. Although the results are sound, presentation and discussion can be improved.

Concerns and comments

  1. In the Abstract and also in the Introduction, I suggest to focus on “camel trypanosomoses” rather than “animals” in general.
  2. It’s not clear if authors examined “All camel samples (388) from the five regions (Line32)”, or calculate the samples size (L67), or “about 388 samples were examined” (L144). Confusing.
  3. In the Abstract state clearly what were found, rather writing primers names.
  4. In the Abstract authors wrote about “seropositive” camels but did not discuss it elsewhere in the manuscript e.g. Methods and Results.
  5. Do not start sentences with abbreviations e.g. “T. evansi”. Correct throughout the ms.
  6. Mention wide range of T. evansi hosts.
  7. I suggest to mention zoonotic potential of T. evansi. See https://pubmed.ncbi.nlm.nih.gov/31881926/
  8. As shown in Figure 2 of the article https://pubmed.ncbi.nlm.nih.gov/35183235/ , T. vivax has been recently reported in Iran. I suggest to state “not testing for T. vivax” as a limitation since T. evansi/T. vivax mixed infections could potentially be present in Oman. See https://pubmed.ncbi.nlm.nih.gov/34506947/
  9. Read a newly published paper on diagnosis of animal trypanosomoses https://pubmed.ncbi.nlm.nih.gov/35183235/ . It will help you to write L47-55 better.
  10. In Methods section write about Trypanosoma species your primers could detect, rather than names of primers. Table 2 of https://www.oie.int/app/uploads/2021/06/compendiumstandarddiagnosticprotocolsanimaltrypansomosesafricanorigin-en.pdf can be helpful.
  11. Words “trypanosome” and “trypanomosis/oses” must not start with capital letters. Correct throughout the ms.
  12. Correct “T. evansi evansi” as “T. evansi” throughout the ms.
  13. L209-212: good to discuss about camel trypanosomis in Africa but also bring camel literature from neighboring Iran.
  14. Manuscript must be revised for English language. Examples of mistakes: L238, L239, 250.
  15. Binomial/scientific names must be typed in italics in References.
  16. Ethical considerations (approval?) must be stated.

Author Response

Dear Reviewer,

all suggestions are considered, except the comment on the lines 88-114 and figure 4.

  • Lines 88-114: it is difficult to reduce this part because the annealing temperature is different in each primers
  • Figure 4: the program I use does not have different colors, I think the map is clear in the manuscript)

Reviewer 2 Report

The manuscript entitled “Detection of Trypanosomoses in Dromedary Camels by using different diagnostic techniques in Northern Oman” reports a survey on detection of Trypanosoma in Dromedary Camels in five regions of Northern of Oman.

In principle, the manuscript presents original results with an average in significance of contents and scientific soundness in regions were the health dromedary camels are important for local economy. The research design is appropriate, with the methods are adequately described. However the quality of the presentation is not satisfactory and the article needs a minor revision before being published on Animals Journal.

  • I suggest to change “Trypanosomoses” with “Trypanosoma infection” in the title. Trypanosomoses suggests a clinical survey which was not performed in the study.
  • Line 33: Authors used “seropositive”, serological tests on whole blood were also performed? If not done, delete this word.
  • Lines 38-39: reconsider these sentences by giving more emphasis to the conclusions.
  • Keywords: add a word related to molecular techniques.
  • Introduction: Add more information about the molecular techniques.
  • Lines 62-64: the meaning of the numbers in brackets is unclear, reconsider the whole paragraph for greater clarity
  • Lines 70-71: This information are available? It might be interesting to consult them as supplementary files.
  • Line 85: “purified” or extracted? Add “as suggested by manufacturer” or similar sentence.
  • Lines 88-114: in order to make this part lightened and if no changes have been made with respect to the protocols mentioned, the authors could consider deleting the details of the PCR (mix and thermal cycle)
  • Lines 108-109: reconsider the grammar in this sentence, please.
  • Lines 115-126: summarize this section, please.
  • Line 128: complete the name of kit, please.
  • Line 147: change the verb in the past
  • Line 144: about?
  • Line 189: “Eight positive samples were sequenced” in the line 166 authors reported seven… I found the part of the results difficult to understand and unclear. Make it clearer, please.
  • Figure 4: I suggest to invert the colours of the map.
  • Line 202: T. evans in italics, please.
  • Line 208: remove the round bracket, please.
  • Lines 209-210: rewrite these sentences, please.
  • Lines 2011: Jordon?
  • Conclusion: Line 248, the authors not reported data about age and sex, this sentence should be modified. Line 250: I suggest to change “I” with “We” or “the authors”.
  • I suggest to rewrite the conclusion, this part should be supported by the results.
  • In conclusion, I found many formatting and punctuation errors in the text. I suggest authors be more careful before submitting a manuscript to an international peer reviewed journal.

Author Response

Dear Reviewers, all suggestions are considered. In detail,

  • Lines 88-114: in order to make this part lightened and if no changes have been made with respect to the protocols mentioned, the authors could consider deleting the details of the PCR (mix and thermal cycle) (it is difficult to reduce this part because the annealing temperature is different in each primers)
  • Figure 4: I suggest to invert the colors of the map.(the program I use not have different colors , I think the map is clear in the manuscript)
